# Alzheimer's disease diagnosis from diffusion tensor images using convolutional neural networks

**Eman N. Marzban**[1]*, **Ayman M. Eldeib**[1], **Inas A. Yassine**[1], **Yasser M. Kadah**[1,2], **for the Alzheimer's Disease Neurodegenerative Initiative**[¶]

**1** Biomedical Engineering and Systems, Faculty of Engineering, Cairo University, Giza, Egypt, **2** Biomedical Engineering Program, Electrical and Computer Engineering Department, King Abdulaziz University, Jeddah, Saudi Arabia

¶ Data used in preparation of this article were obtained from the Alzheimer's Disease Neuroimaging Initiative (ADNI) database (adni.loni.usc.edu). As such, the investigators within the ADNI contributed to the design and implementation of ADNI and/or provided data but did not participate in analysis or writing of this report. A complete listing of ADNI investigators can be found at: http://adni.loni.usc.edu/wp-content/uploads/how_to_apply/ADNI_Acknowledgement_List.pdf

* eman.marzban@eng1.cu.edu.eg

**Data Availability Statement:** The data set is owned by a third-party organization; the Alzheimer's Disease Neuroimaging Initiative

## Abstract

Machine learning algorithms are currently being implemented in an escalating manner to classify and/or predict the onset of some neurodegenerative diseases; including Alzheimer's Disease (AD); this could be attributed to the fact of the abundance of data and powerful computers. The objective of this work was to deliver a robust classification system for AD and Mild Cognitive Impairment (MCI) against healthy controls (HC) in a low-cost network in terms of shallow architecture and processing. In this study, the dataset included was down-loaded from the Alzheimer's disease neuroimaging initiative (ADNI). The classification methodology implemented was the convolutional neural network (CNN), where the diffusion maps, and gray-matter (GM) volumes were the input images. The number of scans included was 185, 106, and 115 for HC, MCI and AD respectively. Ten-fold cross-validation scheme was adopted and the stacked mean diffusivity (MD) and GM volume produced an AUC of 0.94 and 0.84, an accuracy of 93.5% and 79.6%, a sensitivity of 92.5% and 62.7%, and a specificity of 93.9% and 89% for AD/HC and MCI/HC classification respectively. This work elucidates the impact of incorporating data from different imaging modalities; i.e. structural Magnetic Resonance Imaging (MRI) and Diffusion Tensor Imaging (DTI), where deep learning was employed for the aim of classification. To the best of our knowledge, this is the first study assessing the impact of having more than one scan per subject and propose the proper maneuver to confirm the robustness of the system. The results were competitive among the existing literature, which paves the way for improving medications that could slow down the progress of the AD or prevent it.

(ADNI). Data are publicly and freely available from the http://adni.loni.usc.edu/data-samples/access-data/ Institutional Data Access / Ethics Committee (contact via http://adni.loni.usc.edu/data-samples/access-data/) upon sending a request that includes the proposed analysis and the named lead investigator.    Further, please find more details about the ADNI project and data acquisition and sharing policies and protocol: Data sharing policy and data access process: http://adni.loni.usc.edu/wp-content/uploads/how_to_apply/ADNI_DSP_Policy.pdf ADNI steering committee and list of acknowledgement for publications using ADNI repository: http://adni.loni.usc.edu/wp-content/uploads/how_to_apply/ADNI_Acknowledgement_List.pdf ADNI protocol and ethics statement: http://adni.loni.usc.edu/wp-content/themes/freshnews-dev-v2/documents/clinical/ADNI-2_Protocol.pdf

**Funding:** The authors received no funding for this work.

**Competing interests:** The authors have no competing interests to declare.

## Introduction

Neurodegenerative diseases have gained increasing attention in the past few decades; these include Alzheimer's Disease (AD), Mild Cognitive Impairment (MCI), and others. Several research groups have tackled the usage of machine learning algorithms for the sake of detection, localization, prediction of the disease, or clustering different diseases or disease stages [1–3]. Image based diagnosis of AD is important and required mainly to avoid subjective assessments [4]. Deep learning-based methods gives successful results particularly in medical image analysis [5] due to flexible and efficient formulations [6].

In 2017, over 121 thousand people died from AD, in the United States, making it the sixth leading cause of death. Between the years 2000 and 2017, the number of deaths due to AD has increased by 145% [7]. By 2050, the number of people older than 60 years will be increased by 1.25 billion, equivalent to 22% of the global population, with 79% living in the world's less developed countries [8]. The annual expenses for the disease per is around $868 and $3,109 per person in low-income and lower-to middle- income countries respectively [9].

MCI can be an antecedent to several neurodegenerative diseases [1,10]. Prominently, MCI is considered to be the prodromal phase to AD [7,11]. Around 15–20% of people elder than 65 years were diagnosed with MCI because of different pathologies. In a two-year follow-up, 15% of the subjects with MCI would develop dementia, and in a five-year follow-up 32% of subjects, with MCI, would develop AD [7].

Several research projects proposed machine learning algorithms assessing AD with different goals based on different types of features, including; Cerebrospinal Fluid (CSF) histopathology, cognitive questionnaire tests, and Medical Imaging, such as; Magnetic Resonance Imaging (MRI) and Diffusion Tensor Imaging (DTI) [1,12–17].

In this work, the objective was to classify AD and MCI from healthy controls (HC) using a convolutional neural network (CNN), where DTI and MRI were employed. Moreover, all diffusion maps were investigated and compared; Mean Diffusivity (MD), Fractional Anisotropy (FA), and Mode of Anisotropy (MO). Moreover, the effect of the time interval between two subsequent scans was investigated to assess the convenient period between one's scans to avoid overfitting.

## Materials and methods

The dataset, employed in this study, is owned by a third-party organization; the Alzheimer's Disease Neuroimaging Initiative (ADNI). A complete description of ADNI and up-to-date information is available at http://adni.loni.usc.edu/ and data access requests are to be sent to http://adni.loni.usc.edu/data-samples/access-data/. Detailed inclusion criteria for the diagnostic categories can be found at the ADNI website (http://adni.loni.usc.edu/methods, ADNI2 manual page 27). All ADNI studies are conducted according to the Good Clinical Practice guidelines, the Declaration of Helsinki, and U.S. 21 CFR Part 50 (Protection of Human Subjects), and Part 56 (Institutional Review Boards). Written informed consent was obtained from all participants before protocol-specific procedures were performed. The ADNI protocol was approved by the Institutional Review Boards of all of the participating institutions. The ethics committees/institutional review board that approved the ADNI study are listed within S1 File. The dataset employed is formed of 406 subjects: 185, 106, and 115 subjects with HC, MCI and AD respectively. The subjects' characteristics are listed in Table 1.

The preprocessing of the scans was adopted from the pipeline introduced in [19,20]. MRI $T_1$ scans were spatially segmented and normalized to the Montreal Neurological Institute (MNI) template using the Statistical Parametric Mapping (SPM12) software; specifically, the Computational Anatomy Toolbox (CAT12) was utilized and the Diffeomorphic Anatomical

**Table 1. Subjects' characteristics.**

|  | Male/Female (no. unique) | No. scans separated at least a year or more | Age | MMSE | Years of Education |
|---|---|---|---|---|---|
| **HC (n = 185)** | 89/96 (55) | 110 | 73.6±6.1 | 29.0±1.2 | 16.4±2.7 |
| **MCI (n = 106)** | 66/40 (44) | 59 | 73.3±5.8 | 26.8±1.9 | 16.3±2.6 |
| **AD (n = 115)** | 69/46 (50) | 57 | 75.7±8.1 | 23.0±2.5 | 15.5±3.0 |

HC: Healthy controls, MCI: Mild cognitive impairment, AD: Alzheimer's disease, MMSE: Mini-mental state exam[18]

Registration using Exponentiated Lie algebra (DARTEL) algorithm was implemented [21]. Linear regression was implemented to remove the effect of the Total Intracranial Volume (TIV) [22]. This pipeline output several files; such as the White Matter (WM) volume, Gray Matter (GM) volume, TIV values, and the deformation fields (to and from the MNI space). On the other hand, the DTI scans were preprocessed as per the guidelines of the FMRIB Software Library (FSL) [23]; where the eddy currents were corrected, the skull was stripped, the diffusion tensor was calculated, and the diffusion maps were calculated. Last, the DTI maps were co-registered with the normalized $T_1$ scans of the same subject at the same time point via the SPM coregister toolbox [24].

Three main maps of diffusion can be calculated from DTI, named; MD, FA, and MO [25,26]. MD is the average of the eigenvalues of the diffusion tensor ellipsoid [27]. FA is a measure of the flow in the axons being isotropic or closer to anisotropic (0 is perfect isotropy and 1 is perfect anisotropy). MO reflects the skewness of the flow; i.e. is it closer to tubal (-1), spherical (0), or planar (1) flow. In neurodegenerative diseases, including Alzheimer's disease, the demyelination can be perceived as an increase in Radial Diffusivity (RD) and a decrease in FA [28,29].

The hippocampus and the entorhinal cortex are the main and earliest regions that develop anatomical atrophy in the case of AD [26,28,30–36]. Thus, the bounding box, including the hippocampus and the entorhinal cortex, was identified via the Harvard-Oxford [37–40] and the Juelich [41] atlases respectively; originally, the scans were 121×145×121 and by selecting the Volume of Interest (VOI), they became 61×37×38 (Fig 1).

To address the classification task, a 2D CNN was employed, since it has the advantage of taking into consideration the spatial relationships between pixels; especially with a pathology that would progress over time and brain regions [19,42–44].

The proposed CNN consisted of the image input layer, convolutional filters as described below, batch normalization layer, ReLU layer [45], maxpooling layer, fully connected layer,

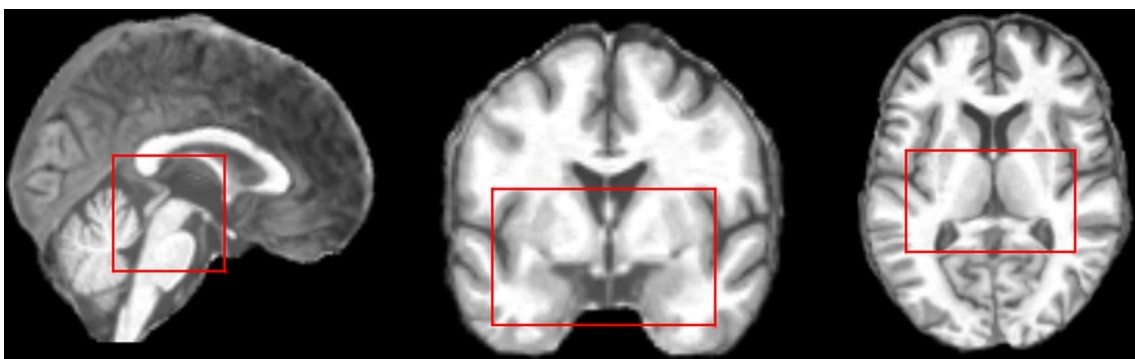

**Fig 1. The hippocampus and the entorhinal cortex bounding box.**

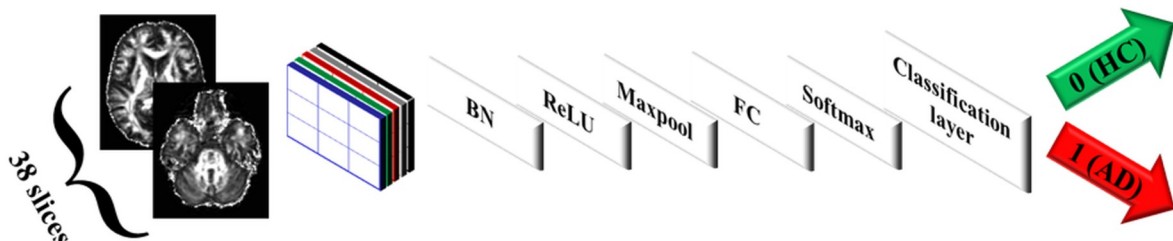

**Fig 2. CNN architecture (FA maps as an example of input scans).** BN: Batch normalization, FC: Fully connected ayer.

softmax layer, and the classification layer (Fig 2). The weights of the network were calculated using the gradient descent optimization using the Root Mean Square Propagation (rmsprop) algorithm [46]. Recently, Sobolev gradient based optimization has been used in deep network based methods to diagnose AD [47,48]. However, the standard gradient descent optimization is efficient in the proposed approach in terms of computation.

The concept of 1×1 convolution was first introduced by Lin et al. [49], whereas its usage has been scarce in medical applications [50,51]. Its role in decreasing the complexity while increasing the nonlinearities -hence, the discriminative ability- was later clarified in [52].

In order to select the optimal network hyperparameters such as the network depth, filters' size and number, iterative experiments were employed where one layer was added and its filter size was optimized before adding the next layer. The optimal sizes were selected when the highest performance measures were met. It is worth noting that the learning of the weights was done via mini-batch scheme; where the batch size, the learning rate, and the number of epochs were 20, 0.001 and 60 respectively. The ten-fold cross-validation implies 10% test set and 90% training set; further, the training set is split into 75% for the network training, and 25% for the validation of the parameters. Validation was performed once per epoch.

In this work, three experiments were performed:

- **Analysis of individual and cascaded maps:** The MD volume was fed to CNN and the performance measures of the test set were calculated; the same was done for FA and MO and the optimal CNN parameters were selected. In addition, the three diffusion volumes were cascaded and fed to the CNN and the optimal CNN parameters were selected for the cascaded volume. The same setting was done for the cascaded MD and GM volumes. Cascading in this study was done by concatenating the diffusion map volumes following each other in depth. Thus, the original size of 61×37×38 for one map would increase to 61×37×76 and 61×37×114 in case of two and three maps respectively; where the third dimension is normal to the axial plane as described in Fig 2

- **Analysis while including a single scan per year for the same subject:** The impact of excluding temporally-close scans (less than a year); i.e. portions vs. annual was assessed in this text. In particular, the dataset comprised subjects who have been scanned more than once; the interval between two subsequent scans was not fixed. Thus, all scans were explored altogether (denoted in this study by *portions*). In addition, scans that remained after excluding those belonging to the same subjects but scanned within less than a year; either from preceding or succeeding scan (denoted by *annual*) were explored as well. In other words, if subject X has scans $X_{S1}$, $X_{S2}$, and $X_{S3}$ sorted by the date the scan was performed; where $X_{S1}$ and $X_{S2}$ were taken within less than a year, and $X_{S2}$ and $X_{S3}$ were taken within a year or more, only $X_{S1}$ and $X_{S3}$ would be kept for that subject X; which is referred as *annual*. Whereas, if all

scans for X were retained irrespective of the interval; i.e. keeping $X_{S1}$, $X_{S2}$, and $X_{S3}$, it would be referred as *portions*.

- **Analysis of segregated versus mixed training and test datasets**: Separating the cross-validation folds by IDs or random assignment to any fold; segregated vs. mixed was evaluated in this text. In this analysis, the impact of multiple scans per subject was exploited in two ways; to put all scans belonging to one subject in either a training or test set per cross-validation which was referred to previously as "segregated", or just to randomize the selection of scans per cross-validation irrespective of the ID of the subject which was referred as "mixed".

Five performance measures were calculated; Area Under the Curve (AUC), accuracy, sensitivity, specificity, and F1- measure[53]. Since the MD mean of the performance measures was primarily the highest in comparison with the other maps, a statistically significant difference between all other maps and MD was analyzed. The Sign test was employed [54,55] since the population was not always normal, assessed by the Shapiro-Wilk test [56], and there were only ten points of observations (number of the cross-validation folds).

To plot the Receiver Operating Characteristic (ROC) curve for any experiment, ten-fold cross validation passes, having different x-y pairs (sensitivity/TPR and 1-specificity/FPR) corresponding to each cross-validation pass, were utilized. The ten curves were interpolated to a common x-axis, named False Positive Rate (FPR), and calculated the average for the other axis, named True Positive Rate (TPR).

For each fold in the cross-validation, the ROC curve has a set of FPR and TPR points forming the curve. First, a common arbitrary set of FPR values was chosen. In order to calculate the average ROC curve of the ten folds, the (FPR, TPR) pairs were sorted in a monotonically increasing fashion with respect to FPR. All the ten curves were looped over, where each loop was unique over the FPR values. To avoid the problem of multiple TPR-values for the same FPR value; i.e. vertical lines, the (FPR, TPR) pair were selected at the last value of FPR (corresponding to the largest value of TPR denoted by $TPR_{max}$ for the same value of FPR). Then, the (FPR, $TPR_{max}$) was interpolated to the previously-selected common FPR-grid. The same method was applied for the rest of the curves, such that all of them coincide on the same grid of FPR, then the average was calculated.

The aim of this work was to provide an automatic classification of the MCI and AD versus HC. For some subjects having multiple scans at different timepoints, the effect of selecting only scans that were taken a year or more from the previous one with respect to the same subject was investigated. In addition, the impact of having different timepoint scans for the same subject in the training set and how the separation based on the subject is assessed in terms of the effect on the overall performance.

The implementation was done on a 64-bit Windows server 2019 machine, Intel Xeon CPU E5-2650 @ 2 GHz processor, eight cores, and 384 GB RAM. The CNN architecture was built using MATLAB ver. R2018b.

## Results

In this study, several objectives were addressed for detecting AD via a machine learning technique; namely CNN. the first objective is to search for the best values for the CNN hyperparameters that would maximize performance. Whereas, the second objective is study if the diffusion maps would yield a good discrimination between different classes or fusion with other structural data will boost the performance. The third objective is to evaluate the impact of the time gap between two successive scans belonging to the same subject. Finally, the study

was interested in assessing the effect of mixing the training and test sets or segregating them such that all scans belonging to the same subject are in either the training set or the test set.

Upon evaluating the different hyperparameters of 2D CNNs, the optimal CNN size, for one volume (MD, MO, FA, or GM) each of which is 61×37×38, is formed of one layer in depth having five filters each of which was 5×5×38. On the other hand, the optimal CNN size for cascaded volumes experiments; namely MD+MO+FA of size 61×37×114 and GM+MD of size 61×37×76, is formed of two layers; where the first included thirty 1×1×114 or thirty 1×1×76 filters respectively and the second layer included five 3×3×30 filters. Regarding 2D CNNs, it is worth pointing out that the depth of the filters must match with that of the input volumes, and that the depth of the output of the convolution must match with the number of filters [57].

### Analysis of individual and cascaded maps

Regarding the maps themselves, the MD maps were roughly statistically significant than other diffusion maps, in comparison, and also the three volumes cascaded (Tables 2 and 3 respectively). MD maps resulted in a classification accuracy of 88.9% and 71.1%, a sensitivity of 83.5% and 51.9%, a specificity of 91.7% and 81.8% and AUC of 0.93 and 0.68 for classifying AD and MCI respectively from HC. In the experiments implemented in this work, the FA yielded better results than MO. FA resulted in an accuracy of 86% and 72.1%, a sensitivity of 78.7% and 50%, a specificity of 90.1% and 79.4%, and AUC of 0.88 and 0.73 for classifying AD and MCI respectively from HC. On the other hand, MO resulted in an accuracy of 82.8% and 64.4%, a sensitivity of 73.8% and 37.4%, a specificity of 87.9% and 79.4%, and an AUC of 0.88 and 0.62 for HC/AD and HC/MCI classification respectively. Feeding the GM to the proposed CNN improved the results but not significantly (Table 2). GM resulted in an accuracy of 91.3% and 75.7%, a sensitivity of 88.3% and 60.7%, and a specificity of 92.8% and 84% and an AUC of 0.96 and 0.80, for classifying AD and MCI respectively from HC.

Further, incorporating the GM with the MD (cascading them as deeper volume denoted by MD+GM) improved the results (Table 3), sometimes significantly depending on the performance measure involved, compared with either MD or GM alone. Specifically, MD+GM produced an accuracy of 93.5% and 79.6%, a sensitivity of 92.5% and 62.7%, a specificity of 93.9% and 89% and an AUC of 0.94 and 0.84 for AD and MCI classification respectively. Cascading the three maps resulted in the least performance (Table 3); MD+MO+FA produced an accuracy of 78.6% and 70.8%, a sensitivity of 66.3% and 41.5%, a specificity of 85.6% and 87.3% and an AUC of 0.86 and 0.74 for the classification of AD and MCI respectively versus HC.

### Analysis while including a single scan per year for the same subject

Generally speaking, excluding the scans that belonged to the same subject that were carried out within less than a year resulted in an insignificant drop in the performance in terms of accuracy, AUC, sensitivity, specificity, and F1-score, as shown in Tables 2 and 3.

### Analysis of segregated versus mixed training and test datasets

Mixing up the scans for one subject in both the training and test sets in one cross-validation yielded overfitting; in particular, the results were generally statistically significantly higher in mixed portions experiments with respect to segregated ones. The level of significance was less when the input scans were removed if two scans for the same subject were performed in less than a year (Table 3). The accuracy and AUC for the cascaded mixed maps were 16.9% and 0.12 respectively *higher* than the corresponding segregated ones for HC/AD classification and 22.2% and 0.25 respectively for HC/MCI classification; this highlights the overfitting severity encountered.

**Table 2. Classification results.**

| | Portions segregated | | | | | Annual segregated | | | | |
|---|---|---|---|---|---|---|---|---|---|---|
| | MD | Mode | FA | GM | MD and GM | MD | Mode | FA | GM | MD and GM |
| **HC/ AD** | | | | | | | | | | |
| **Acc** | 0.889±0.099 (0.898) | 0.828±0.089 (0.830) | 0.860±0.106 (0.895) | 0.913±0.077 (0.913) | 0.935±0.078 (0.965) | 0.868±0.109 (0.910) | 0.814±0.090 (0.789) | 0.826±0.096 (0.879) ‡ | 0.922±0.056 (0.938) | 0.904±0.049 (0.882) |
| **AUC** | 0.931±0.083 (0.969) | 0.878±0.139 (0.910) * | 0.878±0.144 (0.934) | 0.955±0.058 (0.977) | 0.941±0.082 (0.972) | 0.936±0.084 (0.982) | 0.858±0.116 (0.833) | 0.876±0.111 (0.897) | 0.976±0.041 (1.000) */‡‡ | 0.974±0.036 (0.985) |
| **Sens** | 0.835±0.202 (0.905) | 0.738±0.160 (0.773) | 0.787±0.212 (0.818) | 0.883±0.187 (1.000) | 0.925±0.087 (0.955) | 0.857±0.077 (0.833) | 0.643±0.132 (0.667) *** | 0.680±0.150 (0.667) ** | 0.910±0.096 (0.917) | 0.893±0.093 (0.833) |
| **Spec** | 0.917±0.129 (1.000) | 0.879±0.104 (0.889) | 0.901±0.089 (0.889) | 0.928±0.111 (1.000) | 0.939±0.112 (1.000) | 0.873±0.150 (0.909) | 0.900±0.117 (0.955) | 0.900±0.117 (0.909) | 0.927±0.094 (1.000) | 0.909±0.086 (0.909) |
| **F1** | 0.840±0.149 (0.861) | 0.755±0.131 (0.766) | 0.796±0.167 (0.869) | 0.876±0.118 (0.895) * | 0.918±0.092 (0.950) * | 0.827±0.123 (0.861) | 0.704±0.130 (0.641) ** | 0.727±0.144 (0.775) **/‡ | 0.891±0.071 (0.889) | 0.867±0.061 (0.857) ‡ |
| **HC/ MCI** | | | | | | | | | | |
| **Acc** | 0.711±0.170 (0.737) | 0.644±0.154 (0.650) * | 0.721±0.116 (0.719) | 0.757±0.130 (0.754) | 0.796±0.139 (0.776) * | 0.703±0.132 (0.735) | 0.638±0.094 (0.647) | 0.632±0.181 (0.706) * | 0.745±0.115 (0.765) ** | 0.722±0.139 (0.735) |
| **AUC** | 0.681±0.247 (0.697) | 0.619±0.193 (0.587) | 0.732±0.171 (0.740) | 0.800±0.159 (0.860) | 0.842±0.124 (0.852) | 0.644±0.232 (0.583) | 0.730±0.141 (0.705) | 0.648±0.175 (0.652) | 0.745±0.157 (0.758) * | 0.773±0.129 (0.795) |
| **Sens** | 0.519±0.215 (0.500) | 0.374±0.323 (0.286) * | 0.499±0.249 (0.450) | 0.607±0.307 (0.427) * | 0.627±0.298 (0.527) * | 0.490±0.238 (0.500) | 0.207±0.225 (0.167) ** | 0.323±0.168 (0.333) | 0.487±0.194 (0.500) | 0.607±0.169 (0.667) § |
| **Spec** | 0.818±0.224 (0.892) | 0.794±0.308 (0.889) | 0.845±0.114 (0.861) | 0.840±0.124 (0.838) | 0.890±0.120 (0.917) | 0.818±0.148 (0.818) | 0.873±0.172 (0.955) | 0.800±0.218 (0.818) | 0.882±0.122 (0.909) | 0.782±0.232 (0.818) |
| **F1** | 0.564±0.201 (0.563) | 0.389±0.223 (0.353) * | 0.542±0.215 (0.511) | 0.616±0.217 (0.544) | 0.664±0.231 (0.628) */§ | 0.523±0.220 (0.523) | 0.236±0.234 (0.234) ** | 0.399±0.208 (0.500) | 0.563±0.211 (0.606) | 0.609±0.143 (0.606) |

AD: Alzheimer's disease, MCI: mild cognitive impairment, Acc: accuracy, AUC: area under ROC curve, sens: sensitivity, spec: specificity, MD: mean diffusivity, MO: mode of anisotropy, FA: fractional anisotropy, GM: gray-matter volume, numbers are displayed as mean±standard deviation (median)

* Statistically significant from the corresponding *MD* measures using the Sign test p≤0.1

** Statistically significant from the corresponding *MD* measures using the Sign test p≤0.05

*** Statistically significant from the corresponding *MD* measures using the Sign test p≤0.01

‡ Statistically significant from the corresponding *portions* measures using the Sign test p≤0.1

‡‡ Statistically significant from the corresponding *portions* measures using the Sign test p≤0.05

§ Statistically significant from the corresponding *GM* measures using the Sign test p≤0.1

The ROC curves for all analyses are displayed in Fig 3, and summary of results is tabulated in Table 4.

## Discussion

MD outperformed both MO and FA, when employing the individual maps for the classification task with AUC of 0.93, 0.88 and 0.88 for MD, FA, and MO respectively, and this seems to be in concordance with the results from [17,26,29,64]. Douaud et al. [26] reported out that the significant variations in MD, as opposed to FA and MO, primarily in the amygdala-hippocampus complex. Kantarci et al. [29]found out that the MD in the hippocampal and para-hippocampal areas was complementary to the GM volume for the classification of HC/AD. Firbank et al. [66] added that the clusters where MD was significantly higher in AD subjects than that of controls were primarily in the left temporal lobe; that was parallel with atrophy in the grey matter in these locations. Further, Rose et al. [67] reported that MD was elevated significantly at the hippocampus, amygdala, and entorhinal cortex, whereas, FA was reduced significantly mainly at the thalamus. Also, they showed that the cortical areas with increased MD correlate

**Table 3. Classification results of the stacked diffusion maps.**

|  | Portions mixed | Portions segregated | Annual mixed | Annual segregated |
|---|---|---|---|---|
| **HC/ AD** |  |  |  |  |
| **Acc** | 0.955±0.040 (0.965) | 0.786±0.108 (0.776) ** | 0.886±0.103 (0.939) | 0.814±0.114 (0.818) |
| **AUC** | 0.988±0.017 (0.992) | 0.864±0.146 (0.904) **/§ | 0.964±0.054 (0.985) | 0.876±0.119 (0.876) * |
| **Sensitivity** | 0.897±0.102 (0.905) | 0.663±0.196 (0.618) * | 0.757±0.299 (0.833) | 0.663±0.223 (0.667) § |
| **Specificity** | 0.989±0.023 (1.000) | 0.856±0.109 (0.861) *** | 0.955±0.064 (1.000) | 0.891±0.112 (0.909) |
| **F1** | 0.934±0.061 (0.950) | 0.688±0.170 (0.699) ** | 0.775±0.291 (0.909) | 0.700±0.182 (0.641) § |
| **HC/ MCI** |  |  |  |  |
| **Acc** | 0.930±0.033 (0.929) | 0.708±0.104 (0.714) *** | 0.870±0.046 (0.879) | 0.662±0.097 (0.676) *** |
| **AUC** | 0.991±0.007 (0.989) | 0.738±0.146 (0.762) *** | 0.957±0.063 (0.976) | 0.753±0.091 (0.742) *** |
| **Sensitivity** | 0.842±0.096 (0.800) | 0.415±0.288 (0.300) ** | 0.660±0.138 (0.667) | 0.360±0.238 (0.333) *** |
| **Specificity** | 0.978±0.053 (1.000) | 0.873±0.170 (0.946) ** | 0.982±0.038 (1.000) | 0.827±0.163 (0.864) *** |
| **F1** | 0.895±0.050 (0.889) | 0.466±0.208 (0.445) ** | 0.773±0.093 (0.775) | 0.398±0.201 (0.437) *** |

AD: Alzheimer's disease, MCI: mild cognitive impairment, Acc: accuracy, AUC: area under ROC curve, sens: sensitivity, spec: specificity, MD: mean diffusivity, MO: mode of anisotropy, FA: fractional anisotropy, GM: gray-matter volume, numbers are displayed as mean±standard deviation (median)

* Statistically significant from the corresponding *mixed* measures using the Sign test p≤0.1

** Statistically significant from the corresponding *mixed* measures using the Sign test p≤0.05

*** Statistically significant from the corresponding *mixed* measures using the Sign test p≤0.01

§ Statistically significant from the corresponding *MD* measures using the Sign test p≤0.1

with regions of reduced gray matter density measured using structural MRI in patients with AD. It is worthy pointing out that the results in this work, coincide with [67].

In this work, the FA yielded better results than MO. This seems to be in contrast to the low-sample study of [65] where the MO yielded accuracy that was ~7%-10% higher than that driven by FA in HC/AD and HC/MCI respectively. It is worth noting that the sample size, used in this study, was at least five-fold that of Lee et al [65]. The cascaded diffusion maps yielded worse performance, but not always significantly, than employing the MD.

The GM volumes alone, in agreement with the literature, improved the results [17,68]; this is attributed to the fact that AD is prominently characterized by amyloid plaques and neurofi-brillary tangles that deposit in the GM which, in turn, leads to the death of the neurons and the thinning of the cortex or simply atrophy [69–72]. Oishi et al. reported in their study *"DTI is useful for localizing and quantifying the anatomical abnormalities, but apparently not adequate to investigate the histopathological background of the diseases"* [71]. They explained that the DTI measures could be affected by the pathology or other reasons. For example, the diffusion lasts for up to 100 ms in a radius of up to 10 µm that is to be averaged over a voxel of 2–3 mm in size; this indeed makes it more sensitive to the presence of multiple fiber bundles and partial volume effect [71,73]. In addition, in Henf et al.'s work, they concluded that without applying the partial volume correction, MD was not superior to gray matter volume in separating MCI and AD from HC [73].

It is important to assert that in this work, the volumes namely; MD, FA, and MO, and GM and MD were cascaded. Whereas, Wen et al. [62] assessed the MD and FA values over the GM mask (Table 4), and therefore, the performance of the two works cannot be properly compared.

Dyrba et al. [17], using the European DTI study on dementia (EDSD) cohort, reported that combining the MD with GM extracted from structural MRI, where Support Vector Machine (SVM) was utilized, had worsened the results of GM alone. Moreover, the authors reported that the GM utilization outperformed the MD in terms of accuracy, sensitivity and specificity

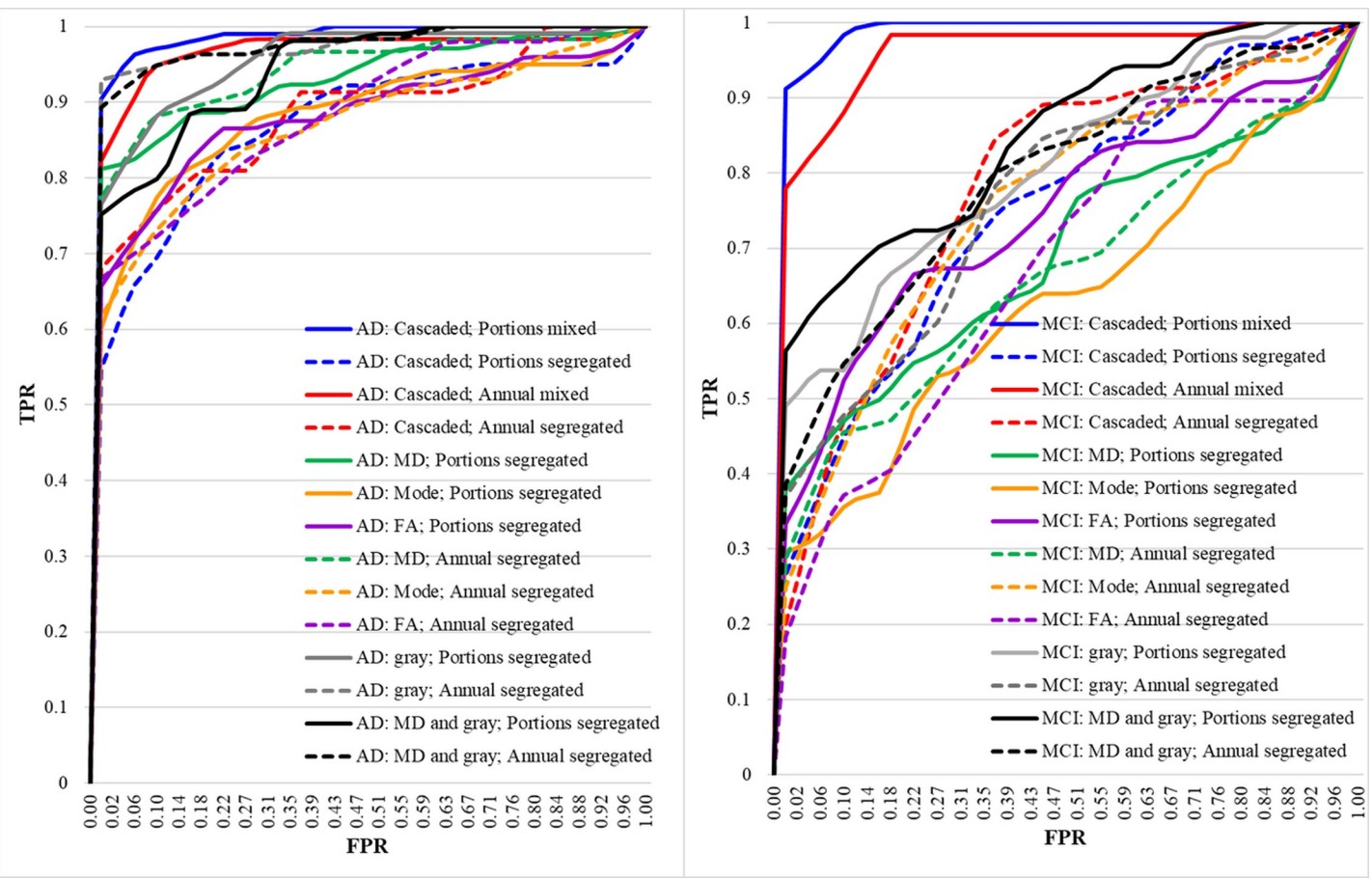

**Fig 3. ROC curves of left: AD/HC classification, right: MCI/HC classification.**

(Table 4). In this work, incorporating the GM with the MD improved the results; not always significantly though.

One of the biggest hurdles encountered when dealing with machine learning in general, and neural networks in specific, is the limitation of the dataset; especially when dealing with medical data; this is the main cause of overfitting [74]. The batch normalization layer is used to reduce the problem of overfitting [75,76] due to its importance in deep learning [77]. In addition, the usage of small-sized filters is usually enhancing the test set performance measures compared to larger filters as explained in the Methods section, through decreasing the overfitting which aligns with Pereira et al. [78] who advocated that small filter sizes of 3×3 would minimize the effect of overfitting since the number of parameters to be learnt decreased. Further, Simonyan and Zisserman [52] explained that the effective receptive field of two stacked 3×3 convolutional layers was equivalent to a single 5×5 layer and that of three stacked 3×3 convolutional layers was equivalent to a single 7×7 layer. Moreover, increasing the number of layers increases nonlinearities, which also decreases the weights, to be optimized, by 77% and 81% for the first and the second case respectively. In addition, the proposed architecture comprised of only one or two layers in depth to alleviate the problem of overfitting; this is in agreement with Ahmed et al. [76] and RStudio online tutorials [79]. Though, ten-fold cross-validation technique was incorporated to give a good estimate about the generalizability of the classification[80,81].

**Table 4. Summary of results.**

| Study | Study sample | Methodology | Results | Data type(s) (Dataset) |
|---|---|---|---|---|
| Liu et al., 2018 [58] | 199 AD and 229 HC | CNN on landmarks learnt by statistical significance tests | Accuracy and AUC: HC/AD: 90.56% and 0.96 | MRI (ADNI) |
| Lin et al., 2018 [59] | 188 AD, 229 HC, 169 MCIc, and 193 MCInc | CNNs | Accuracy and AUC: HC/AD: 88.79% MCIc/MCInc: 79.9% and 0.86 | MRI (ADNI) |
| Islam et al., 2018 [60] | 316 non-demented, 70 very-mild, 28 mild, and 2 with moderate dementia | Ensemble of very deep (~120–169 convolutional layers) CNNs | Average multilabel classification accuracy of 93.18% | MRI (OASIS [61]) |
| Wen et al., 2018 [62] | 46 AD and 46 HC | Linear SVM for MD and FA whole maps only | Accuracy and AUC: MD-GM 76%, 0.83 FA-GM 71%, 0.77 | MRI and DTI (ADNI) |
| Khvostikov et al., 2018 [63] | 48 AD, 108 MCI, and 58 HC | MD only, only hippocampus, CNN, with/without augmentation | Accuracy: MRI, MD HC/AD: 85%, 97% HC/MCI: 66%, 63% MCI/AD: 75%, 80% | MRI and DTI (ADNI) |
| Ahmed et al., 2017 [14] | 45 AD, 58 MCI, and 52 HC | MD, MRI, hippocampus bounding box, multiple kernel learning | Accuracy: HC/AD: 90.2% HC/MCI: 79.42% MCI/AD: 76.63% | MRI, DTI, and CSF (ADNI) |
| Nir et al., 2015 [64] | 37 AD, 113 MCI, and 50 HC | Linear SVM for MD and FA maximum path density (MPD) maps (MD performed better) | Accuracy: HC/AD: 74.5–84.9% HC/MCI: 68.3–79% (MD only) | DTI (ADNI) |
| Lee et al., 2015 [65] | 22 AD, 47 MCI, and 22 HC | SVM on FA and MO from TBSS | Accuracy: HC/AD: FA 90.9%, MO 97.7 MCI/AD: FA 88.4%, MO 98.6 | DTI (ADNI) |
| Dyrba et al, 2012 [17] | 137 AD and 143 HC | Multi-kernel SVM, MD, FA, WM, GM | Accuracy: HC/AD: MD 83.3% FA 80.3% GM 89.3% MD+GM 88.7% MD+GM+FA 89.1% | MRI and DTI (EDSD) |
| The proposed algorithm | 115 AD, 106 MCI, and 185 HC | Small CNN, MD, FA, MO, GM | Accuracy and AUC: HC/AD: MD 88.9%, 0.93 FA 86%, 0.88 MO 82.8%, 0.88 GM 91.3%, 0.96 MD+GM 93.5%, 0.94 HC/ MCI: MD 71.1%, 0.68 FA 72.1%, 0.73 MO 64.4%, 0.62 GM 75.7%, 0.80 MD+GM 79.6%, 0.84 | MRI and DTI (ADNI) |

AD: Alzheimer's disease, MCI: mild cognitive impairment, MCIc: MCI convert, MCInc: MCI non convert, HC: healthy controls, MD: mean diffusivity, MO: mode of anisotropy, FA: fractional anisotropy, GM: gray matter, WM: white matter, CSF: cerebrospinal fluid, TBSS: tract-based spatial statistics, SVM: support vector machine, CNN: convolutional neural network, AUC: area under curve, MRI: magnetic resonance imaging, DTI: diffusion tensor imaging, ADNI: Alzheimer's disease neuroimaging initiative, OASIS: open access series of imaging studies, EDSD: European DTI study on dementia.

It can be noticed that the drop of the performance between portions (all scans of the subject are included) and annual (only scans a year or more apart are included) was minor; this could be attributed to the fact that the number of the scans, upon being annually-scanned, dropped -at least- to half of those without this constraint (Table 1).

As shown previously in the *Results* section, the effect of segregating the scans of the same subject to either the learning or the testing data versus randomly selecting the scans with no constraints during the cross-validation folds that the accuracy and AUC dropped by around

17% and 0.124 respectively at p<0.05 for the HC/AD classification and 22.2% and 0.25 respectively at p<0.01 for HC/MCI classification.

This could be interpreted as an overfitting case where during the cross-validation pass, the network considered the *temporal* instance of the scan of the same subject as a previously seen scan in the training stage; where there is a spatial dependency in the same subject as the disease progresses. This overfitting case would promote the classification performance task. [44,82–84].

Further, the average execution time for the entire ten-fold cross-validation, training and testing, was 12.5 minutes, and the average time per one scan during testing was 0.005 seconds; this is quite competitive when the availability of a graphical processing unit (GPU) is restricted or not possible.

It is important to highlight that all models, proposed in this study, had their specificity higher than their sensitivity; i.e. they are better at handling true negatives than true positives. Coherent to this, some analyses suggested the presence of a trade-off between these two measures[85–87]. This is mainly due to the fact that number of healthy subjects used in this study was quite larger than the number of MCI and AD subjects [85,86].

It is worthy to mention that incorporating the CSF amyloid data could be considered to be interpreted and asses its role in differentiating cognitive deficits. Longitudinal assessment of the cases should be studied; this is a promising means of early detection of the onset of AD which helps aid AD drug discovery and testing.

## Conclusion

In this paper, a CNN was handcrafted to classify MCI and AD from HC. The MD, FA, MO, GM, MD+GM scans were compared; MD was the best-performing diffusion map amongst the diffusion maps regarding classification in terms of accuracy, specificity, and AUC of 88.9%, 91.7% and 0.93 respectively for HC/AD classification, and 71.1%, 81.8% and 0.68 respectively for HC/MCI classification. Combining GM with MD enhanced the performance but below the 5% significance level; to give an accuracy, a specificity, and an AUC of 93.5%, 93.9% and 0.94 respectively for HC/AD classification and 79.6%, 89% and 0.84 respectively for HC/MCI classification.

The dataset comprised more than one instance per subject and in this work, it is recommended that the training and test sets should be split such that one's scans were in the same pile; i.e. the IDs of the subjects in the training set and the test set *should not* overlap.

## Supporting information

**S1 File. This file contains a list of the ethics committees/institutional review boards that approved the ADNI study.**
(PDF)

## Acknowledgments

The authors would like to acknowledge the **Universität Rostock, Germany** for providing us with access to the Information technology and media center (ITMZ) machine. Further, the authors would like to thank **Dr. Martin Dyrba**, the German center for neurodegenerative diseases (DZNE), Rostock, Germany for his guidance in some preprocessing steps. The authors would like to thank the ADNI (http://adni.loni.usc.edu/) and the Functional Imaging in Neuropsychiatric Disorders Lab (http://findlab.stanford.edu/) investigators for publicly sharing their valuable neuroimaging data.

## Author Contributions

**Conceptualization:** Eman N. Marzban, Ayman M. Eldeib, Inas A. Yassine.

**Data curation:** Eman N. Marzban.

**Formal analysis:** Eman N. Marzban.

**Investigation:** Eman N. Marzban, Inas A. Yassine.

**Methodology:** Eman N. Marzban, Ayman M. Eldeib, Inas A. Yassine.

**Project administration:** Ayman M. Eldeib, Inas A. Yassine, Yasser M. Kadah.

**Software:** Eman N. Marzban.

**Supervision:** Ayman M. Eldeib, Inas A. Yassine, Yasser M. Kadah.

**Validation:** Eman N. Marzban.

**Visualization:** Eman N. Marzban.

**Writing – original draft:** Eman N. Marzban.

**Writing – review & editing:** Eman N. Marzban, Inas A. Yassine, Yasser M. Kadah.

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
