## [Decision Letter · Decision Letter 0]

14 Feb 2020

PONE-D-19-35687

Alzheimer’s disease diagnosis from diffusion tensor images using convolutional neural networks

PLOS ONE

Dear Dr. Marzban,

Thank you for submitting your manuscript to PLOS ONE. After careful consideration by two Reviewers and an Academic Editor, please make the suggested corrections posed by both Reviewers so I can render a decision on this manuscript.

**Comments to the Author**

1. Is the manuscript technically sound, and do the data support the conclusions?

Reviewer #1: Yes

Reviewer #2: Yes

2. Has the statistical analysis been performed appropriately and rigorously? 

Reviewer #1: Yes

Reviewer #2: Yes

3. Have the authors made all data underlying the findings in their manuscript fully available?

Reviewer #1: Yes

Reviewer #2: No

4. Is the manuscript presented in an intelligible fashion and written in standard English?

Reviewer #1: Yes

Reviewer #2: Yes

5. Review Comments to the Author

Reviewer #1: The aim of this work is to provide an automatic classification of the MCI and AD versus HC.

For some subjects having multiple scans at different timepoints, the effect of selecting only scans that were taken a year or more from the previous one with respect to the same subject has been investigated.

In addition, the impact of having different timepoint scans for the same subject in the training set and how the separation based on the subject has been assessed in terms of the effect on the overall performance.

The topic handled in this study is important and an active research area.

The paper has been well-organized.

However, the following revisions are required to improve its quality.

1) Optimization is an important stage and the standard (Euclidean) gradient descent optimization has been used in this work.

The authors should add the following statement to make clear that this approach presents efficient results, so it has been applied:

"Recently, Sobolev gradient based optimization has been used in deep network based methods to diagnose AD [R1,R2]. However, the standard gradient descent optimization is efficient in the proposed approach in terms of computation."

R1:"Diagnosis of Alzheimer's Disease with Sobolev Gradient Based Optimization and 3D Convolutional Neural Network", Numerical Methods in Biomedical Engineering, 35(7),2019

R2:"Fully Automated Classification of Brain Tumors Using Capsules for Alzheimer’s Disease Diagnosis", IET Image Processing, 10.1049/iet-ipr.2019.0312, 2019

2) The following sentence should be updated to make its meaning clearer:

"In comparison with the diffusion maps in comparison, MD outperformed the other two maps; namely: FA and MO."

3) There are many different approaches in this area.

The reason to use a deep learning based technique in this study should be explained clearly by supporting appropriate recent studies.

Therefore, the following statements can be added:

"Image based diagnosis of AD is important and required mainly to avoid subjective assessments [R1]. Deep learning based methods gives successful results particularly in medical image analysis [R2] due to flexible and efficient formulations [R3]."

R1: "Biomedical Information Technology: Image Based Computer Aided Diagnosis Systems", The 7th Int.Conf. on Advanced Technologies, 2018

R2: "Deep Learning in Medical Image Analysis: Recent Advances and Future Trends", Int. Conf. Computer Graphics, Visualization, Computer Vision and Image Processing 2017 (CGVCVIP 2017), Lisbon, Portugal

R3:"Formulas Behind Deep Learning Success",Int.Conf. on Applied Analysis and Mathematical Modeling (ICAAMM2018), 2018

4) There is typo in; "Moreover, Increasing the number ....", which should be "Moreover, increasing the number...."

5) The meaning of the sentence;

"....the limitation of the dataset; especially when dealing with medical data; this is the main cause of overfitting."

should be supported with the following recent work indicating main issues in deep learning based methods:

"Challenges and Recent Solutions for Image Segmentation in the Era of Deep Learning", 9th Int. Conf. on Image Processing Theory, Tools and Applications (IPTA),2019

The following sentence;

"The batch normalization layer is used to reduce the problem of overfitting [68,69]"

should be updated as

"The batch normalization layer is used to reduce the problem of overfitting [68,69] due to the importance of it in deep learning [R]"

R:"On The Importance of Batch Size for Deep Learning", Int. Conf. on Mathematics (ICOMATH2018), An Istanbul Meeting for World Mathematicians, Istanbul, Turkey,2018

Reviewer #2: The authors compared different input data settings and experiments settings to predict AD/MCI/HC from DTI using CNNs. The experiments and analyses are comprehensive with in-depth discussions. The paper is well presented and has enough novelty. The only improvement I can think of is that the author weren't super clear about the CNNs' structures. It is unclear it is a 2D or a 3D CNN. In addition, the author didn't mention how multiple input were handled when there were more than one input to the CNN, e.g. MD+GM. And the cascading method is also not clearly defined. Lacking these details might lead to difficulty in reproducing the work. I suggest the author add those technical details to the method section.

6. PLOS authors have the option to publish the peer review history of their article (what does this mean?). If published, this will include your full peer review and any attached files.

**Do you want your identity to be public for this peer review?** For information about this choice, including consent withdrawal, please see our Privacy Policy.

Reviewer #1: No

Reviewer #2: No

We would appreciate receiving your revised manuscript by June, 2020. To enhance the reproducibility of your results, we recommend that if applicable you deposit your laboratory protocols in protocols.io, where a protocol can be assigned its own identifier (DOI) such that it can be cited independently in the future. For instructions see: http://journals.plos.org/plosone/s/submission-guidelines#loc-laboratory-protocols

We look forward to receiving your revised manuscript.

Kind regards,

Stephen D. Ginsberg, Ph.D.

Section Editor

PLOS ONE

Journal Requirements:

This study was carried out in accordance with the recommendations of the Declaration of Helsinki. The ADNI protocol was approved by the Institutional Review Boards of all of the participating institutions. Informed written consent was obtained from all participants at each site.

"Data collection and sharing for this project was funded by the

Alzheimer’s Disease Neuroimaging Initiative (ADNI) (National Institutes of Health Grant

U01 AG024904). The ADNI was launched in 2003 by the National Institute on Aging (NIA),

the National Institute of Biomedical Imaging and Bioengineering (NIBIB), the Food and

Drug Administration (FDA), private pharmaceutical companies and non-profit organizations,

as a $60 million, 5-year public-private partnership. ADNI is funded by the National Institute

on Aging, the National Institute of Biomedical Imaging and Bioengineering, and through

generous contributions from the following: Alzheimer’s Association; Alzheimer’s Drug

Discovery Foundation; Araclon Biotech; BioClinica, Inc.; Biogen Idec Inc.; Bristol-Myers

Squibb Company; CereSpir, Inc.; Eisai Inc.; Elan Pharmaceuticals, Inc.; Eli Lilly and

Company; EuroImmun; F. Hoffmann-La Roche Ltd and its affiliated company Genentech,

Inc.; Fujirebio; GE Healthcare; IXICO Ltd.; Janssen Alzheimer Immunotherapy Research &

Development, LLC.; Johnson & Johnson Pharmaceutical Research & Development LLC.;

Lumosity; Lundbeck; Merck & Co., Inc.; Meso Scale Diagnostics, LLC.; NeuroRx Research;

Neurotrack Technologies; Novartis Pharmaceuticals Corporation; Pfizer Inc.; Piramal

Imaging; Servier; Takeda Pharmaceutical Company; and Transition Therapeutics. The

Canadian Institute of Health Research is providing funds to support ADNI clinical sites in

Canada. Private sector contributions are facilitated by the Foundation for the National

Institutes of Health (www.fnih.org). The grantee organization is the Northern California

Institute for Research and Education, and the study is coordinated by the Alzheimer’s Disease

Cooperative Study at the University of California, San Diego. ADNI data are disseminated by

the Laboratory for Neuroimaging at the University of Southern California."

"The authors received no specific funding for this work."

Additionally, because some of your funding information pertains to commercial funding, we ask you to provide an updated Competing Interests statement, declaring all sources of commercial funding.

In your Competing Interests statement, please confirm that your commercial funding does not alter your adherence to PLOS ONE Editorial policies and criteria by including the following statement: "This does not alter our adherence to PLOS ONE policies on sharing data and materials.” as detailed online in our guide for authors  http://journals.plos.org/plosone/s/competing-interests.  If this statement is not true and your adherence to PLOS policies on sharing data and materials is altered, please explain how.

Please include the updated Competing Interests Statement and Funding Statement in your cover letter. We will change the online submission form on your behalf.

---

## [Author Response · Author response to Decision Letter 0]

26 Feb 2020

Dear Dr. Ginsberg, 

We would like to submit the revised manuscript entitled “Alzheimer’s disease diagnosis from diffusion tensor images using convolutional neural networks” for reconsideration by the journal of PLOS ONE.

We have responded carefully to the questions and concerns raised by the reviewers, where our responses and actions items are inline in blue and preceded by ampersand symbol (&). 

Please address all correspondence concerning this manuscript to me at eman.marzban@eng1.cu.edu.eg

Thank you for your consideration of this manuscript. 

Sincerely,

Eman N. Marzban

Faculty of Engineering, Cairo University, Cairo Uni. str., 12613, Giza, Egypt

Tel.: +2-0235703426, +2-01003320868

 Comments to the Author

1. Is the manuscript technically sound, and do the data support the conclusions?

Reviewer #1: Yes

Reviewer #2: Yes

2. Has the statistical analysis been performed appropriately and rigorously? 

Reviewer #1: Yes

Reviewer #2: Yes

3. Have the authors made all data underlying the findings in their manuscript fully available?

Reviewer #1: Yes

Reviewer #2: No

& We updated the cover letter and appended this sentence to it, paragraph no. 3, line 20: 

“Data availability statement: The data set is owned by a third-party organization; the Alzheimer’s Disease Neuroimaging Initiative (ADNI). Data are publicly and freely available from the http://adni.loni.usc.edu/data-samples/access-data/ Institutional Data Access / Ethics Committee (contact via http://adni.loni.usc.edu/data-samples/access-data/) upon sending a request that includes the proposed analysis and the named lead investigator.”________________________________________

4. Is the manuscript presented in an intelligible fashion and written in standard English?

Reviewer #1: Yes

Reviewer #2: Yes

5. Review Comments to the Author

Reviewer #1: The aim of this work is to provide an automatic classification of the MCI and AD versus HC.

For some subjects having multiple scans at different timepoints, the effect of selecting only scans that were taken a year or more from the previous one with respect to the same subject has been investigated.

In addition, the impact of having different timepoint scans for the same subject in the training set and how the separation based on the subject has been assessed in terms of the effect on the overall performance.

The topic handled in this study is important and an active research area.

The paper has been well-organized.

However, the following revisions are required to improve its quality.

1) Optimization is an important stage and the standard (Euclidean) gradient descent optimization has been used in this work.

The authors should add the following statement to make clear that this approach presents efficient results, so it has been applied:

"Recently, Sobolev gradient based optimization has been used in deep network based methods to diagnose AD [R1,R2]. However, the standard gradient descent optimization is efficient in the proposed approach in terms of computation."

R1:"Diagnosis of Alzheimer's Disease with Sobolev Gradient Based Optimization and 3D Convolutional Neural Network", Numerical Methods in Biomedical Engineering, 35(7),2019

R2:"Fully Automated Classification of Brain Tumors Using Capsules for Alzheimer’s Disease Diagnosis", IET Image Processing, 10.1049/iet-ipr.2019.0312, 2019

& We would like to thank the reviewer for his valuable suggestions.

The references were added as suggested as references No. 47 and no. 48, in page 31. The added references are cited in line 122, page 8.

2) The following sentence should be updated to make its meaning clearer:

"In comparison with the diffusion maps in comparison, MD outperformed the other two maps; namely: FA and MO."

& The sentence was changed to be “MD outperformed both MO and FA, when employing the individual maps for the classification task with AUC of 0.93, 0.88 and 0.88 for MD, FA, and MO respectively.” in line 286, page 21.

3) There are many different approaches in this area.

The reason to use a deep learning based technique in this study should be explained clearly by supporting appropriate recent studies.

Therefore, the following statements can be added:

"Image based diagnosis of AD is important and required mainly to avoid subjective assessments [R1]. Deep learning based methods gives successful results particularly in medical image analysis [R2] due to flexible and efficient formulations [R3]."

R1: "Biomedical Information Technology: Image Based Computer Aided Diagnosis Systems", The 7th Int.Conf. on Advanced Technologies, 2018

R2: "Deep Learning in Medical Image Analysis: Recent Advances and Future Trends", Int. Conf. Computer Graphics, Visualization, Computer Vision and Image Processing 2017 (CGVCVIP 2017), Lisbon, Portugal

R3:"Formulas Behind Deep Learning Success",Int.Conf. on Applied Analysis and Mathematical Modeling (ICAAMM2018), 2018

& The references were added as suggested, as references No. 4, 5, and 6, in page 28. These references are cited in lines 48 and 49, page 4.

4) There is typo in; "Moreover, Increasing the number ....", which should be "Moreover, increasing the number...."

& The typo has been updated to " Moreover, increasing the number of layers increases nonlinearities,…” in line 337 page 23.

5) The meaning of the sentence;

"....the limitation of the dataset; especially when dealing with medical data; this is the main cause of overfitting."

should be supported with the following recent work indicating main issues in deep learning based methods:

"Challenges and Recent Solutions for Image Segmentation in the Era of Deep Learning", 9th Int. Conf. on Image Processing Theory, Tools and Applications (IPTA),2019

& The reference was added as suggested, as reference No. 74 in page 33. The reference is cited in line 329 page 22. 

The following sentence;

"The batch normalization layer is used to reduce the problem of overfitting [68,69]"

should be updated as

"The batch normalization layer is used to reduce the problem of overfitting [68,69] due to the importance of it in deep learning [R]"

R:"On The Importance of Batch Size for Deep Learning", Int. Conf. on Mathematics (ICOMATH2018), An Istanbul Meeting for World Mathematicians, Istanbul, Turkey,2018

& The reference was added as suggested, as reference No 77 in page 33. The reference is cited in line 330 in page 22. 

Reviewer #2: The authors compared different input data settings and experiments settings to predict AD/MCI/HC from DTI using CNNs. The experiments and analyses are comprehensive with in-depth discussions. The paper is well presented and has enough novelty. The only improvement I can think of is that the author weren't super clear about the CNNs' structures. It is unclear it is a 2D or a 3D CNN. In addition, the author didn't mention how multiple input were handled when there were more than one input to the CNN, e.g. MD+GM. And the cascading method is also not clearly defined. Lacking these details might lead to difficulty in reproducing the work. I suggest the author add those technical details to the method section.

& We would like to thank the reviewer for the valuable comment and inquiries. We have updated the description of the network, in several places, to include all missing information. The updated paragraphs are listed below:

1- Paragraph No. 1, starting at line 114 in page 8:

"To address the classification task, a 2D CNN was employed, having the advantage of taking into consideration the spatial relationships between pixels; especially with a pathology that would progress over time and brain regions [19,42–44]."

2- Paragraph No 2, line 146 in page 9:

 "Cascading in this study was done by concatenating the diffusion map volumes following each other in depth. Thus, the original size of 61×37×38 for one map would increase to 61×37×76 and 61×37×114 in case of two and three maps respectively; where the third dimension is normal to the axial plane as described in Fig 2."

3- Paragraph No 2, line 211 in page 11:

"Upon evaluating the different hyperparameters of 2D CNNs, the optimal CNN size, for one volume (MD, MO, FA, or GM) each of which is 61×37×38, is formed of one layer in depth having five filters each of which was 5×5×38. On the other hand, the optimal CNN size for cascaded volumes experiment; namely MD+MO+FA of size 61×37×114 and GM+MD of size 61×37×76, is formed of two layers; where the first layer included thirty 1×1×114 or thirty 1×1×76 filters respectively and the second layer included five 3×3×30 filters. Regarding 2D CNNs, it is worth pointing out that the depth of the filters must match with that of the input volumes, and that the depth of the output of the convolution must match with the number of filters [57]."________________________________________

6. PLOS authors have the option to publish the peer review history of their article (what does this mean?). If published, this will include your full peer review and any attached files.

Do you want your identity to be public for this peer review? For information about this choice, including consent withdrawal, please see our Privacy Policy.

Reviewer #1: No

Reviewer #2: No

& All figures were confirmed to comply with your standards via figures@plos.org

We would appreciate receiving your revised manuscript by June, 2020. To enhance the reproducibility of your results, we recommend that if applicable you deposit your laboratory protocols in protocols.io, where a protocol can be assigned its own identifier (DOI) such that it can be cited independently in the future. For instructions see: http://journals.plos.org/plosone/s/submission-guidelines#loc-laboratory-protocols

• A rebuttal letter that responds to each point raised by the academic editor and reviewer(s). This letter should be uploaded as separate file and labeled 'Response to Reviewers'.

• A marked-up copy of your manuscript that highlights changes made to the original version. This file should be uploaded as separate file and labeled 'Revised Manuscript with Track Changes'.

• An unmarked version of your revised paper without tracked changes. This file should be uploaded as separate file and labeled 'Manuscript'.

We look forward to receiving your revised manuscript.

Kind regards,

Stephen D. Ginsberg, Ph.D.

Section Editor

PLOS ONE

Journal Requirements:

& Thank you for your valuable comment. The manuscript has been updated to follow the PLOS ONE's style requirements.

This study was carried out in accordance with the recommendations of the Declaration of Helsinki. The ADNI protocol was approved by the Institutional Review Boards of all of the participating institutions. Informed written consent was obtained from all participants at each site.

& Thank you very much for your review. It is important to highlight that we did not implement the data acquisition ourselves; our study was carried out on the publicly and freely available data from the ADNI repository. Where the process is usually started by submitting a research protocol to ADNI. Once the proposal is protocol; ADNI would grant the research team the data access, for download, through their website (http://adni.loni.usc.edu/data-samples/access-data/ ). 

It is important to note that the interaction with human beings, of the ADNI data acquisition team, for scanning and diagnosis was conformant with the declaration of Helsinki.

To mitigate the Journal requirements, we updated the cover letter and appended the following sentence, which would clarify that we employed the data only for the analysis but our study did not employ any patient interaction or data acquisition in Different places as listed below:

1- The revised cover letter, paragraph no. 1, page 2, line 25

“Further, please find more details about the ADNI project and data acquisition and sharing policies and protocol:

Data sharing policy and data access process: http://adni.loni.usc.edu/wp-content/uploads/how_to_apply/ADNI_DSP_Policy.pdf

ADNI steering committee and list of acknowledgment for publications using ADNI repository: http://adni.loni.usc.edu/wp-content/uploads/how_to_apply/ADNI_Acknowledgement_List.pdf

ADNI protocol and ethics statement: http://adni.loni.usc.edu/wp-content/themes/freshnews-dev-v2/documents/clinical/ADNI-2_Protocol.pdf”

2- In the revised cover letter, page 1, paragraph 3, line 20

 "Data availability statement: The dataset, employed in this study, is owned by a third-party organization; the Alzheimer’s Disease Neuroimaging Initiative (ADNI). Data are publicly and freely available from the http://adni.loni.usc.edu/data-samples/access-data/ Institutional Data Access / Ethics Committee (contact via http://adni.loni.usc.edu/data-samples/access-data/) upon sending a request that includes the proposed analysis and the named lead investigator."

3- In the ethics statement in the online submission, we included:

“The proposed algorithm uses the ADNI data repository. All ADNI studies are conducted according to the Good Clinical Practice guidelines, the Declaration of Helsinki, and U.S. 21 CFR Part 50 (Protection of Human Subjects), and Part 56 (Institutional Review Boards). Written informed consent was obtained from all participants before protocol-specific procedures were performed. The ADNI protocol was approved by the Institutional Review Boards of all of the participating institutions. Ethics committees/institutional review boards that approved the study are:

Albany Medical Center Committee on Research Involving Human Subjects Institutional Review Board, Boston University Medical Campus and Boston Medical Center Institutional Review Board, Butler Hospital Institutional Review Board, Cleveland Clinic Institutional Review Board, Columbia University Medical Center Institutional Review Board, Duke University Health System Institutional Review Board, Emory Institutional Review Board, Georgetown University Institutional Review Board, Health Sciences Institutional Review Board, Houston Methodist Institutional Review Board, Howard University Office of Regulatory Research Compliance, Icahn School of Medicine at Mount Sinai Program for the Protection of Human Subjects, Indiana University Institutional Review Board, Institutional Review Board of Baylor College of Medicine, Jewish General Hospital Research Ethics Board, Johns Hopkins Medicine Institutional Review Board, Lifespan - Rhode Island Hospital Institutional Review Board, Mayo Clinic Institutional Review Board, Mount Sinai Medical Center Institutional Review Board, Nathan Kline Institute for Psychiatric Research & Rockland Psychiatric Center Institutional Review Board, New York University Langone Medical Center School of Medicine Institutional Review Board, Northwestern University Institutional Review Board, Oregon Health and Science University Institutional Review Board, Partners Human Research Committee Research Ethics, Board Sunnybrook Health Sciences Centre, Roper St. Francis Healthcare Institutional Review Board, Rush University Medical Center Institutional Review Board, St. Joseph's Phoenix Institutional Review Board, Stanford Institutional Review Board, The Ohio State University Institutional Review Board, University Hospitals Cleveland Medical Center Institutional Review Board, University of Alabama Office of the IRB, University of British Columbia Research Ethics Board, University of California Davis Institutional Review Board Administration, University of California Los Angeles Office of the Human Research Protection Program, University of California San Diego Human Research Protections Program, University of California San Francisco Human Research Protection Program, University of Iowa Institutional Review Board, University of Kansas Medical Center Human Subjects Committee, University of Kentucky Medical Institutional Review Board, University of Michigan Medical School Institutional Review Board, University of Pennsylvania Institutional Review Board, University of Pittsburgh Institutional Review Board, University of Rochester Research Subjects Review Board, University of South Florida Institutional Review Board, University of Southern, California Institutional Review Board, UT Southwestern Institution Review Board, VA Long Beach Healthcare System Institutional Review Board, Vanderbilt University Medical Center Institutional Review Board, Wake Forest School of Medicine Institutional Review Board, Washington University School of Medicine Institutional Review Board, Western Institutional Review Board, Western University Health Sciences Research Ethics Board, and Yale University Institutional Review Board.”

4- And in the Methods section, paragraph No 1, page 5, line 77:

"…All ADNI studies are conducted according to the Good Clinical Practice guidelines, the Declaration of Helsinki, and U.S. 21 CFR Part 50 (Protection of Human Subjects), and Part 56 (Institutional Review Boards). Written informed consent was obtained from all participants before protocol-specific procedures were performed.” 

5- And in the footnote of Methods section, page 6:

“Ethics committees/institutional review boards that approved the ADNI study are: Albany Medical Center Committee on Research Involving Human Subjects Institutional Review Board, Boston University Medical Campus and Boston Medical Center Institutional Review Board, Butler Hospital Institutional Review Board, Cleveland Clinic Institutional Review Board, Columbia University Medical Center Institutional Review Board, Duke University Health System Institutional Review Board, Emory Institutional Review Board, Georgetown University Institutional Review Board, Health Sciences Institutional Review Board, Houston Methodist Institutional Review Board, Howard University Office of Regulatory Research Compliance, Icahn School of Medicine at Mount Sinai Program for the Protection of Human Subjects, Indiana University Institutional Review Board, Institutional Review Board of Baylor College of Medicine, Jewish General Hospital Research Ethics Board, Johns Hopkins Medicine Institutional Review Board, Lifespan - Rhode Island Hospital Institutional Review Board, Mayo Clinic Institutional Review Board, Mount Sinai Medical Center Institutional Review Board, Nathan Kline Institute for Psychiatric Research & Rockland Psychiatric Center Institutional Review Board, New York University Langone Medical Center School of Medicine Institutional Review Board, Northwestern University Institutional Review Board, Oregon Health and Science University Institutional Review Board, Partners Human Research Committee Research Ethics, Board Sunnybrook Health Sciences Centre, Roper St. Francis Healthcare Institutional Review Board, Rush University Medical Center Institutional Review Board, St. Joseph's Phoenix Institutional Review Board, Stanford Institutional Review Board, The Ohio State University Institutional Review Board, University Hospitals Cleveland Medical Center Institutional Review Board, University of Alabama Office of the IRB, University of British Columbia Research Ethics Board, University of California Davis Institutional Review Board Administration, University of California Los Angeles Office of the Human Research Protection Program, University of California San Diego Human Research Protections Program, University of California San Francisco Human Research Protection Program, University of Iowa Institutional Review Board, University of Kansas Medical Center Human Subjects Committee, University of Kentucky Medical Institutional Review Board, University of Michigan Medical School Institutional Review Board, University of Pennsylvania Institutional Review Board, University of Pittsburgh Institutional Review Board, University of Rochester Research Subjects Review Board, University of South Florida Institutional Review Board, University of Southern, California Institutional Review Board, UT Southwestern Institution Review Board, VA Long Beach Healthcare System Institutional Review Board, Vanderbilt University Medical Center Institutional Review Board, Wake Forest School of Medicine Institutional Review Board, Washington University School of Medicine Institutional Review Board, Western Institutional Review Board, Western University Health Sciences Research Ethics Board, and Yale University Institutional Review Board.”

& Thank you for your comment. We updated the cover letter, where the following sentence is appended in page 1, paragraph no. 3, line 20:

“Data availability statement: The data set, employed in this study, is owned by a third-party organization; the Alzheimer’s Disease Neuroimaging Initiative (ADNI). Data are publicly and freely available from the http://adni.loni.usc.edu/data-samples/access-data/ Institutional Data Access / Ethics Committee (contact via http://adni.loni.usc.edu/data-samples/access-data/) upon sending a request that includes the proposed analysis and the named lead investigator.”

"Data collection and sharing for this project was funded by the

Alzheimer’s Disease Neuroimaging Initiative (ADNI) (National Institutes of Health Grant

U01 AG024904). The ADNI was launched in 2003 by the National Institute on Aging (NIA),

the National Institute of Biomedical Imaging and Bioengineering (NIBIB), the Food and

Drug Administration (FDA), private pharmaceutical companies and non-profit organizations,

as a $60 million, 5-year public-private partnership. ADNI is funded by the National Institute

on Aging, the National Institute of Biomedical Imaging and Bioengineering, and through

generous contributions from the following: Alzheimer’s Association; Alzheimer’s Drug

Discovery Foundation; Araclon Biotech; BioClinica, Inc.; Biogen Idec Inc.; Bristol-Myers

Squibb Company; CereSpir, Inc.; Eisai Inc.; Elan Pharmaceuticals, Inc.; Eli Lilly and

Company; EuroImmun; F. Hoffmann-La Roche Ltd and its affiliated company Genentech,

Inc.; Fujirebio; GE Healthcare; IXICO Ltd.; Janssen Alzheimer Immunotherapy Research &

Development, LLC.; Johnson & Johnson Pharmaceutical Research & Development LLC.;

Lumosity; Lundbeck; Merck & Co., Inc.; Meso Scale Diagnostics, LLC.; NeuroRx Research;

Neurotrack Technologies; Novartis Pharmaceuticals Corporation; Pfizer Inc.; Piramal

Imaging; Servier; Takeda Pharmaceutical Company; and Transition Therapeutics. The

Canadian Institute of Health Research is providing funds to support ADNI clinical sites in

Canada. Private sector contributions are facilitated by the Foundation for the National

Institutes of Health (www.fnih.org). The grantee organization is the Northern California

Institute for Research and Education, and the study is coordinated by the Alzheimer’s Disease

Cooperative Study at the University of California, San Diego. ADNI data are disseminated by

the Laboratory for Neuroimaging at the University of Southern California."

"The authors received no specific funding for this work."

Additionally, because some of your funding information pertains to commercial funding, we ask you to provide an updated Competing Interests statement, declaring all sources of commercial funding.

In your Competing Interests statement, please confirm that your commercial funding does not alter your adherence to PLOS ONE Editorial policies and criteria by including the following statement: "This does not alter our adherence to PLOS ONE policies on sharing data and materials.” as detailed online in our guide for authors http://journals.plos.org/plosone/s/competing-interests. If this statement is not true and your adherence to PLOS policies on sharing data and materials is altered, please explain how.

Please include the updated Competing Interests Statement and Funding Statement in your cover letter. We will change the online submission form on your behalf.

& Indeed, we did not receive any fund (neither from research entities nor from commercial ones) in the very strict sense, and we did not implement data acquisition on our own. Nevertheless, we employed the ADNI dataset in our study. It is important to note that it is mandatory to acknowledge the ADNI data repository as per their data sharing policy (the link is listed at the end of the paragraph). ADNI gets the aforementioned fund as an entirely different entity than ours. ADNI is currently an ongoing project where entity itself is funded to perform the data acquisition including the scans, diagnosis, cognitive assessment, other medical maneuvers, and also to disseminate the data upon requests as explained in http://adni.loni.usc.edu/wp-content/uploads/how_to_apply/ADNI_DSP_Policy.pdf.

In order to mitigate the confusion, we added the paragraph below in the revised cover letter in page 2, paragraph no. 1, line 25:

“Further, please find more details about the ADNI project and data acquisition and sharing policies and protocol:

Data sharing policy and data access process: http://adni.loni.usc.edu/wp-content/uploads/how_to_apply/ADNI_DSP_Policy.pdf

ADNI steering committee and list of acknowledgment for publications using ADNI repository: http://adni.loni.usc.edu/wp-content/uploads/how_to_apply/ADNI_Acknowledgement_List.pdf

ADNI protocol and ethics statement: http://adni.loni.usc.edu/wp-content/themes/freshnews-dev-v2/documents/clinical/ADNI-2_Protocol.pdf

Funding Statement: The authors received no specific funding for this work. 

Competing Interests Statement: The authors declare that the research was conducted in the absence of any commercial or financial relationships that could be construed as a potential conflict of interest.”

---

## [Editor Report · Decision Letter 1]

2 Mar 2020

Alzheimer’s disease diagnosis from diffusion tensor images using convolutional neural networks

PONE-D-19-35687R1

Dear Dr. Marzban,

We are pleased to inform you that your manuscript has been judged scientifically suitable for publication and will be formally accepted for publication once it complies with all outstanding technical requirements.

With kind regards,

Stephen D. Ginsberg, Ph.D.

Section Editor

PLOS ONE

---

## [Editor Report · Acceptance letter]

11 Mar 2020

PONE-D-19-35687R1 

Alzheimer’s disease diagnosis from diffusion tensor images using convolutional neural networks 

Dear Dr. Marzban:

I am pleased to inform you that your manuscript has been deemed suitable for publication in PLOS ONE. Congratulations! Your manuscript is now with our production department. 

With kind regards,

on behalf of

Dr. Stephen D. Ginsberg 

Section Editor

PLOS ONE